# Remember how to use it: Effector-dependent modulation of spatial working memory activity in posterior parietal cortex

**Artur Pilacinski** [1,2]☯*, **Melanie S. Höller-Wallscheid**[1], **Axel Lindner**[1,3,4]☯*

**1** Department of Cognitive Neurology, Hertie-Institute for Clinical Brain Research, Tübingen, Germany, **2** Faculty of Psychology and Educational Sciences, University of Coimbra, Coimbra, Portugal, **3** Division of Neuropsychology, Hertie-Institute for Clinical Brain Research, Tübingen, Germany, **4** Department of Psychiatry and Psychotherapy, University Hospital Tübingen, Tübingen, Germany

☯ These authors contributed equally to this work.
* art.pilacinski@gmail.com (AP); a.lindner@medizin.uni-tuebingen.de (AL)

**Data Availability Statement:** All relevant data are within the paper and its Supporting Information files.

## Abstract

Working memory (WM) is the key process linking perception to action. Several lines of research have, accordingly, highlighted WM's engagement in sensori-motor associations between retrospective stimuli and future behavior. Using human fMRI we investigated whether prior information about the effector used to respond in a WM task would have an impact on the way the same sensory stimulus is maintained in memory despite a behavioral response could not be readily planned. We focused on WM-related activity in posterior parietal cortex during the maintenance of spatial items for a subsequent match-to-sample comparison, which was reported either with a verbal or with a manual response. We expected WM activity to be higher for manual response trials, because of posterior parietal cortex's engagement in both spatial WM and hand movement preparation. Increased fMRI activity for manual response trials in bilateral anterior intraparietal sulcus confirmed our expectations. These results imply that the maintenance of sensory material in WM is optimized for motor context, i.e. for the effector that will be relevant in the upcoming behavioral responses.

## Introduction

Working memory (WM) is the key cognitive process that allows bridging between previously encountered sensory information and future action. Yet, the detailed WM processing architecture and its underlying neuronal substrates still remain somewhat elusive [1, 2]. The traditional view on working memory architecture, which builds on Baddeley's and Hitch's multicomponent model, thereby assumes that the sensory information is maintained within domain-specific modules, namely the visuomotor sketchpad, the phonological loop, or the episodic buffer [3]. Brain imaging studies partially support this assumption, showing that WM-processing of a specific content engages the sites of its respective cortical representation, such as fusiform face area for faces, visual cortex for other visual forms, and temporal cortex for

**Funding:** This work was supported by grants from the German Center for Neurodegenerative Diseases, the DFG (CIN; DFG FA361/26-1; Open Access Publishing Fund University of Tübingen), and the BMBF (FKZ 01GQ1002). The funders had no role in study design, data collection and analysis, decision to publish, or preparation of the manuscript. The funding institutions provided salary to Artur Pilacinski (PhD stipend), Axel Lindner (scientist position) and Melanie Hoeller-Wallscheid (PhD stipend) and contributed to the financing of experimental hardware, consumables (subject reimbursement), and publication costs. The specific roles of these authors are articulated in the 'author contributions' section.

**Competing interests:** There are no patents, products in development or marketed products to declare. This does not alter our adherence to PLOS ONE policies on sharing data and materials.

auditory content, [4–7]. These content-specific WM "storages" possess independent processing capacities, further supporting the notion of separate storage modules for stimulus maintenance [8, 9].

While the classical concept of separate memory storage modules relates to the retrospective aspect of WM, i.e. it focuses on the content of the information that has been received, one should keep in mind that the information is maintained within WM in order to be used for future action. This latter, *prospective* aspect of WM, namely the link of WM to motor behavior, has been at the center of various investigations [10–14]. For example, rather than merely storing *retrospective* stimulus material (such as briefly presented visual targets), it has been shown that subjects do form and maintain prospective plans for actions in WM whenever possible (such as a motor plan to look or to reach towards these targets) [11, 14, 15].

Fig 1A and 1B schematically illustrate typical experimental settings which highlight the maintenance of retrospective vs. prospective information in WM. Fig 1A represents a "classical" working memory task (such as the Sternberg task [16]), which usually does not allow for action preparation during WM maintenance. Accordingly, only retrospective stimulus content should be maintained in WM. Fig 1B) represents a task in which the task rule additionally allows a subject to readily plan and maintain an action. Accordingly, it might be the prospective action plan that is (additionally) maintained in WM in such situation. Experimental

## A) "Rule does not define action"

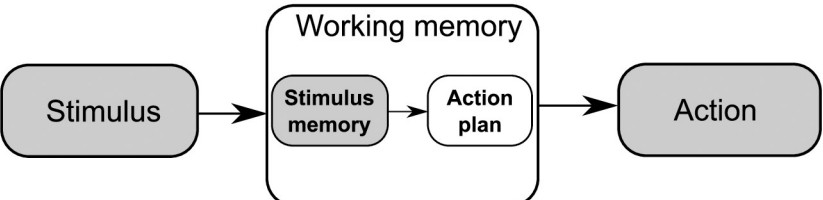

## B) "Rule defines action"

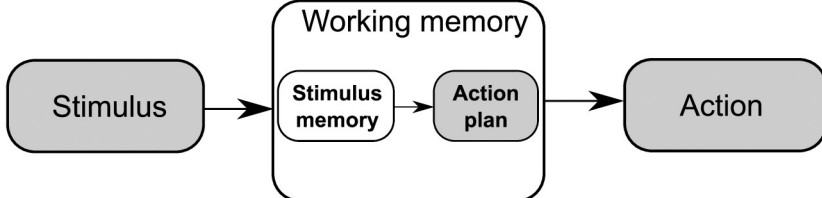

## C) "Rule defines motor context but not action"

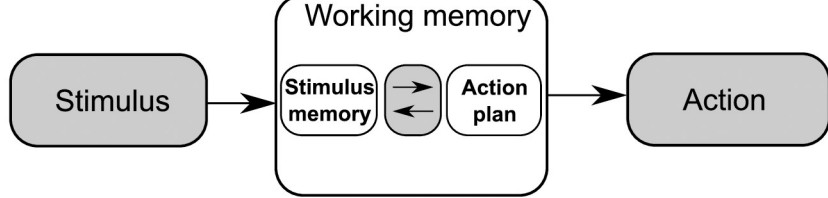

**Fig 1.** Hypothetical influence of varying task rules (A-C) on the maintenance of information in working memory. For further explanation, please refer to the main text.

support has been provided in support of this conceptual distinction between retrospective and prospective information maintenance: Studies of Curtis and colleagues [11, 17] and Lindner and coworkers [14] report differences in brain activity whenever spatial objects not only had to be remembered but, at the same time, were targets to which saccades [11] or reaches [14] should be directed. Yet, in spite of these reported differences, both working memory and action planning tasks recruit an almost identical set of posterior parietal and frontal areas, reflecting strong functional ties between the two aspects of the perception-to-action processing stream [2, 11, 14, 18].

This tangled relation between prospective and retrospective processes mediated by WM is not surprising. In natural environments the retrospective memory content typically predicts the motor context in which the memorized information will be used, as both aspects are naturally linked. For example, in a natural environment the probability that visuospatial positional information (e.g. about the location of an object) will be useful for actions (e.g. for looking or reaching) is usually much higher than for using the same information for speech. Could it be that WM content is maintained in a way that is optimized for the type of action that the memorized information affords (e.g. in a format that is optimized for guiding any upcoming eye and hand movements in case of visuospatial information, etc.)? In other words, could it be that sensory information is not merely stored in its original sensory format (Fig 1A), but (also) adapted to the effector that most likely will make use of this information (Fig 1C), and even if the ultimate action cannot be readily planned (such as in Fig 1B)?

To answer this question, we devised a WM experiment where subjects' brain activity was measured by means of functional magnetic resonance imaging (fMRI). Our experimental task was designed in a way that allowed us to assess such effector-dependent changes in brain activity representing WM maintenance of the same visuospatial stimuli (Fig 1C). Importantly, our design also made sure that an action could not be preplanned based on the sensory material that had to be memorized (compare below for further details). We expected that the obtained BOLD-signals reflecting working memory maintenance should be influenced by effector modality through sensorimotor processes interrelating the retrospective stimulus memory and prospective motor actions (Fig 1C). These sensorimotor processes include the specification of the effector, which is afforded by the stimulus/task, as the main distinctive feature of the upcoming motor program [19]. As already elaborated above, effector specification could, in turn, enable an optimization of memory storage to account for the expected/instructed future use of the memorized sensory material, following the expected sensory-to-motor transformation-chain as far as possible (Fig 1C).

More specifically, we expected that the maintenance of visuospatial information in WM should lead to increased levels of fMRI activity in posterior parietal cortex (PPC) whenever the WM information will be probed with a hand response as compared to a verbal response. This is for a number of reasons. First, PPC is engaged in the sensorimotor transformations of visuospatial information for eye- and hand movements [20]. Second, PPC has been demonstrated to be engaged both in visuospatial working memory tasks as well as in visuospatial planning tasks that engage eye and hand movements [11, 14, 21]. Third, PPC is less engaged in verbal preparation than frontal speech areas [22] and, moreover, it contributes significantly less to verbal as compared to spatial memory tasks [6]. Therefore, if WM maintenance of visuospatial information would be influenced by motor context (i.e. the instructed effector), we expect larger pools of active PPC neurons (and therefore stronger fMRI-activity) in case of hand than in case of verbal responses and–to stress this again–even if the ultimate action plan cannot be readily planned. In other words, if there is any influence of motor context on the maintenance of retrospective sensory information in WM, we should be able to trace it as an effector-dependent BOLD-signal modulation in PPC.

## Results

Our experiment was comprised of two basic paradigms: a working memory, match-to-sample task (M2ST) and a control task (CT). The first one was a classical working memory paradigm with delayed response, where subjects were required to remember a visuospatial pattern of either two or otherwise six circles on the screen (presentation time: 3 sec). Then, after a 14/15 sec delay, a second pattern with the same number of circles came up and subjects had to indicate whether the pattern had changed or not ("same" vs. "different"). Crucially, in each trial we instructed the subjects immediately before the presentation of the circle stimuli to answer either manually (by using a button box held in their right hand) or verbally (compare: Fig 2; see "Materials and methods" for details). This instruction allowed us to manipulate the effector modality, which was relevant for the later behavioral response, on a trial by trial basis in both the M2ST and the CT. In the control task (CT) we also manipulated the instructed effector, but the subjects' role was simply to judge the visual symmetry of circles presented after the delay (using the same responses as in M2ST, i.e. "same" and "different" to denote "symmetric" and "asymmetric", respectively). Hence, the CT instructed the same effectors as in the M2ST but, importantly, did not require subjects to maintain spatial information in WM throughout

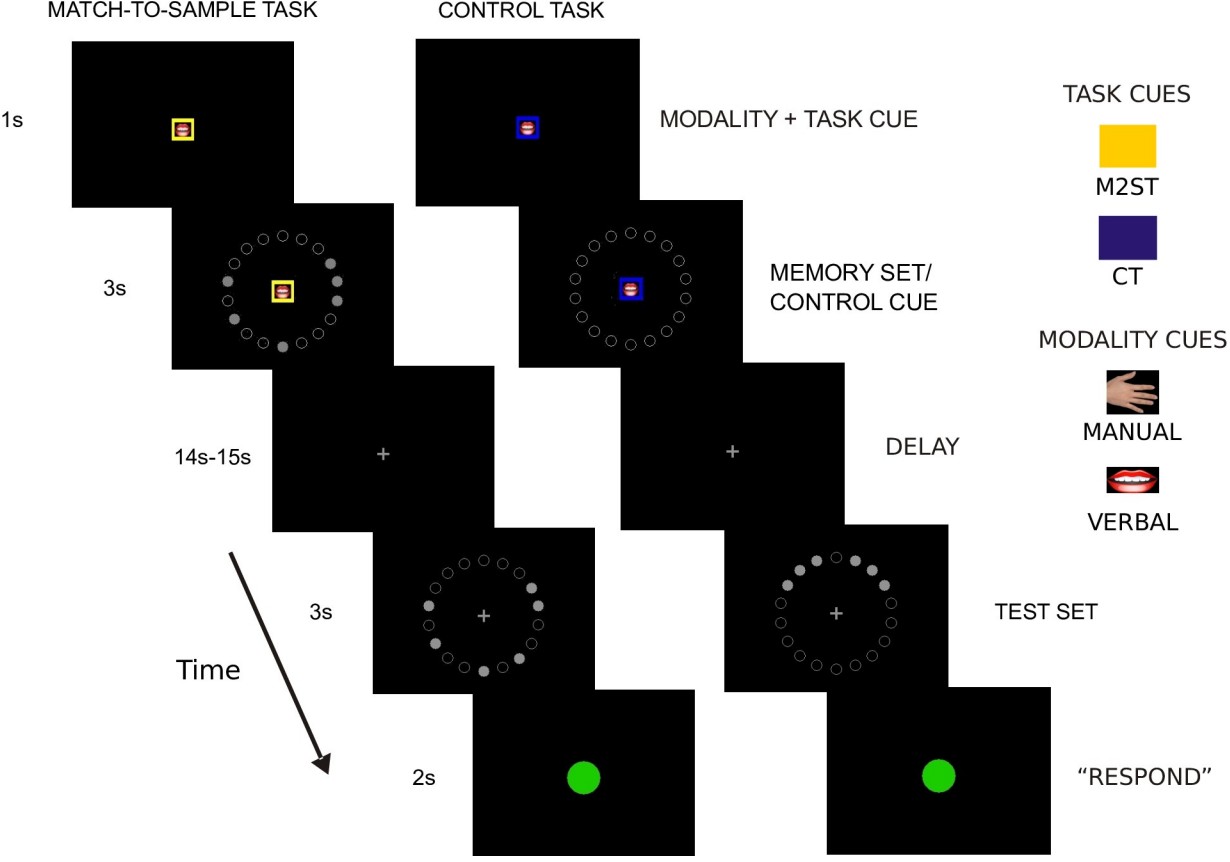

**Fig 2. Exemplary timelines for the working memory task (M2ST) and the control task (CT).** Each trial started with a color cue and an icon symbolizing the current task (M2ST or CT) and the response modality (Manual or Verbal), respectively. After displaying a memory item set (in M2ST) or, alternatively, an irrelevant control cue (in CT), the memory delay began. After the delay, a test set appeared, defining subjects motor response. In the M2ST subjects were asked to determine whether the presented test set matched the earlier memory set. In the CT subjects were asked to indicate whether or not the test set pattern was symmetric. A green "go" cue instructed the onset of the response period, in which subjects had to provide their answers in the correct response modality (manual or verbal). Please note that the visual mask screens were removed from the timeline for clarity. See text for further details.

the delay. The control task thus allowed us to account for any delay-related brain activity in the M2ST that would be merely related to anticipatory preparation of an effector as compared to a true reorganization of WM maintenance through the effector instruction. It is important to stress, again, that our experimental task ensured independence between the visuospatial memory content (i.e. the memory cue locations) and the required motor respons (i.e. binary button presses), which was only specified after the WM maintenance period.

Fourteen volunteers took part in the experiment. All of them were right handed, had no history of neurological disease and had normal or corrected to normal vision (see "Materials and methods" for details). All volunteers gave their written informed consent prior to the experiment, the experimental procedures were carried out in accordance with the declaration of Helsinki and the study was approved by the ethics committee at the University Clinic and the Medical Faculty of the University of Tübingen. Imaging was performed using a 3-Tesla MR-scanner and a twelve channel head coil (Siemens TRIO, Erlangen, Germany). Analyses were performed using SPM8 (Wellcome Center for Neuroimaging), R (R Foundation for Scientific Computing) and custom Matlab (MathWorks) routines.

We first analyzed the subjects' task performance using a 2x2x2 ANOVA with the factors "task" (M2ST vs. CT),"response modality" (manual vs. verbal), and"load" (2 vs. 6 circles). Analyses of hit rates performed across all conditions revealed a significantly higher accuracy in the control task as compared to the memory task (see Fig 3; main effect of task: df = 13, F = 20.73, p = 0.00054, $eta^2_G$ = 0.13).All other effects were not significant (Load: df = 13, F = 0.47, p = 0.5, $eta^2_G$ = 0.004; Response modality: df = 13, F = 1.15, p = 0.3, $eta^2_G$ = 0.008; Response modality x Task: df = 13, F = 0.7, p = 0.4, $eta^2_G$ = 0.005; Load x Task: df = 13, F = 2.99, p = 0.11, $eta^2_G$ = 0.026; Response modality x Load: df = 13, F = 02, p = .9, $eta^2_G < 0.001$; "Task x Load x Response modality": df = 13, F = 0.21, p = 0.65, $eta^2_G < 0.001$).

Next, we quantified brain activity during the delay period of our tasks. To this end we used SPM 8, in which we specified a GLM for each subject: Each of our eight conditions (2 tasks [M2ST & CT] x 2 response modalities [verbal & manual] x 2 'loads' [2 and 6 circles]) was modeled as a separate boxcar regressor of respective duration for both the delay period as well as for the response phase. We modelled the cue-period (+mask) accordingly but only considered two regressors representing our two principle tasks (M2ST & CT). All aforementioned regressors were convolved with the canonical haemodynamic response function in SPM8. Fixation epochs weren't modeled explicitly and served as an implicit baseline. Note that we modeled

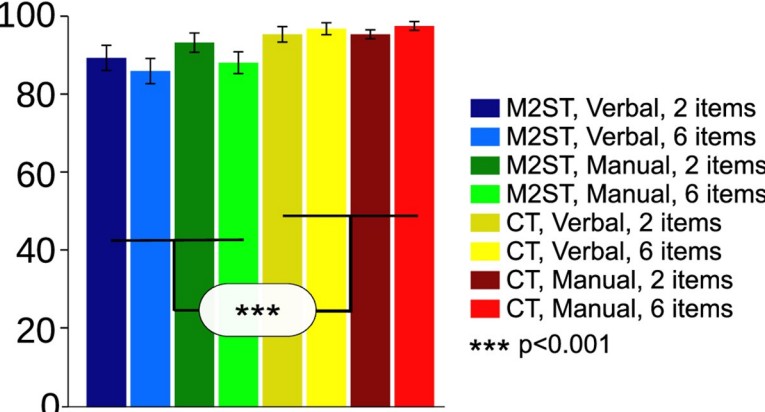

**Fig 3. Task performance.** Across-subject average hit-rates (% correct) in the manual and the verbal response modality and separately for each load (2 or 6 items) and for each task (M2ST and CT). Error bars denote standard error. Hit rates differed significantly between M2ST and CT (for detailed statistics, please refer to the Results section).

the factor 'load' also in the control condition (as was defined by the number of symmetry test items visible in the response phase) despite this variable could not have any influence on brain activity in the preceding delay period.

Fig 4 shows fMRI-activity increases during the delay phase in the M2ST as compared to the CT as revealed by an individual GLM-analysis in a representative subject (A) and by a second-level group-analysis (B). Such signal increases between tasks are expected due to the additional WM processes in M2ST as compared to CT. As we hypothesized, the M2ST yielded higher delay-related fMRI-activity in a number of posterior parietal areas that are typically recruited in both working memory and action planning tasks [14]. These PPC areas were bilateral superior parietal lobule (SPL) and bilateral anterior intraparietal sulcus (aIPS). Our contrast further exhibited several frontal areas, namely bilateral dorsal premotor cortex (PMd), bilateral ventral premotor cortex (PMv), bilateral dorsolateral prefrontal cortex (DLFPC) and supplementary motor area (SMA).

All aforementioned areas in PPC were treated as regions-of-interest (ROIs) in our subsequent analyses. In these ROI analyses we probed for an influence of the instructed effector on delay-related activity in the M2ST as compared to the CT. Please note that we expected to reveal such influence within posterior parietal cortex, as was laid out in detail in the introduction. Yet, to provide a more detailed overview, we also provide explanatory analyses of all other WM-related areas mentioned above (compare S1 Fig) (see "Materials and Methods" for further details and see S1 Table for MNI coordinates of all relevant areas).

From each ROI we extracted the beta weights of all delay-related GLM regressors in each individual. Beta values were expressed in terms of % signal change by dividing them through the overall residual beta estimates (i.e. our implicit baseline measure; compare above). The across-subject averages of these 'normalized' beta weights are depicted in Fig 5 along with the corresponding time-courses of fMRI-activity). Visual inspection reveals the expected activity pattern: First, there are higher levels of activity in the M2ST as compared to the CT (as was the defining criterion for ROI selection). To recapitulate, this was expected due to a contribution of PPC to visuospatial WM. Second, and more importantly, the delay-related activity was higher for instructed hand than for verbal responses in the M2ST but not in the CT. Finally, there is a clear load effect present in the M2ST (but not in CT).

To statistically analyze the beta weights, which seemingly matched our experimental hypothesis at first glance, we performed 2x2x2 ANOVAs (factors "Task", "Load", and "Response Modality"). This allowed us to directly compare delay-related activity in M2ST vs. CT as a function of effector modality and, thereby, to control for any unspecific motor readiness common to both tasks [23, 24]. If our hypothesis were true, we'd expect to reveal an interaction between "Task" and "Response Modality". In addition, we expected an interaction between "Task" and "Load", as there should be a WM load-effect in M2ST but not in CT.

Indeed, our analyses confirmed a significant "Task x Load" interaction in all of our parietal ROIs (SPL left: df = 13, F = 6.38, p = 0.025, $eta^2_G$ = 0.03; SPL right: df = 13, F = 6.54, p = 0.024, $eta^2_G$ = 0.04; aIPS left: df = 13, F = 8.58, p = 0.012, $eta^2_G$ = 0.02; aIPS right: df = 13, F = 12.6, p = 0.0036, $eta^2_G$ = 0.03). Most importantly, we further revealed the effector-dependent differences in fMRI-activity between M2ST and CT in area aIPS bilaterally ("Response Modality"x"-Task". aIPS left: df = 13, F = 8.06, p = 0.013, $eta^2_G$ = 0.03; aIPS right: df = 13, F = 10.14, p = 0.0072, $eta^2_G$ = 0.02) (compare Table 1 and S2 Table for a full overview over all results).

The table shows the results of our 2x2x2 repeated measures ANOVAs with the factors "Response Modality" (manual vs. verbal), "Load" (2 vs. 6 items) and "Task" (match-to-sample task vs. control task), which was calculated across delay-related betas extracted from each parietal ROI in each individual.

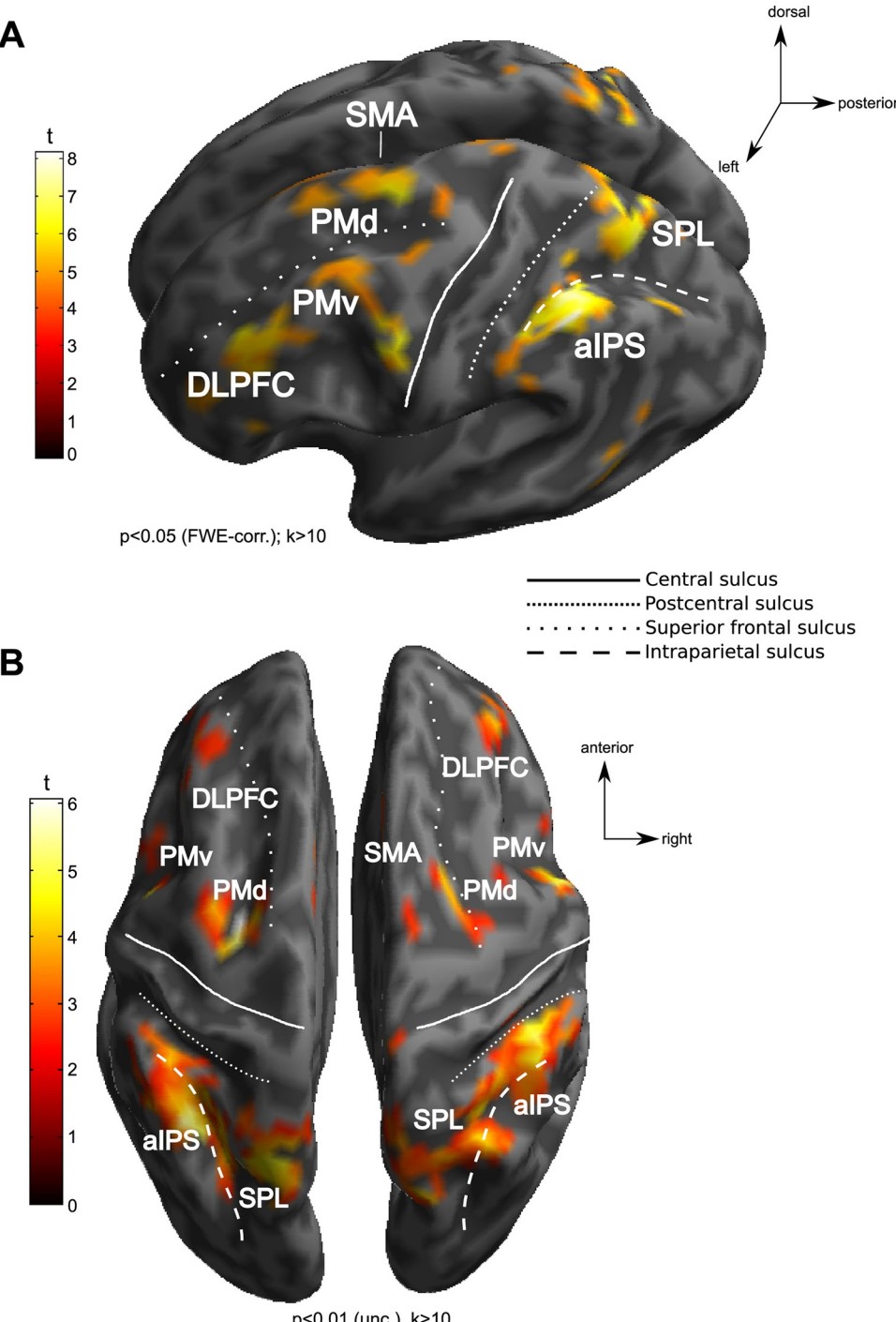

**Fig 4. Working memory areas.** The figures depict WM areas exhibiting stronger activity in M2ST than in CT during the delay period. A) depicts an exemplary subject's WM contrast map. Such individual maps were used to functionally define our WM ROIs. B) depicts the WM contrast map of a corresponding random-effects group analysis.

## Discussion

In our study we observed that changes in working-memory-related BOLD activity in anterior IPS were influenced by the effector used for subjects' subsequent response. This effect was not

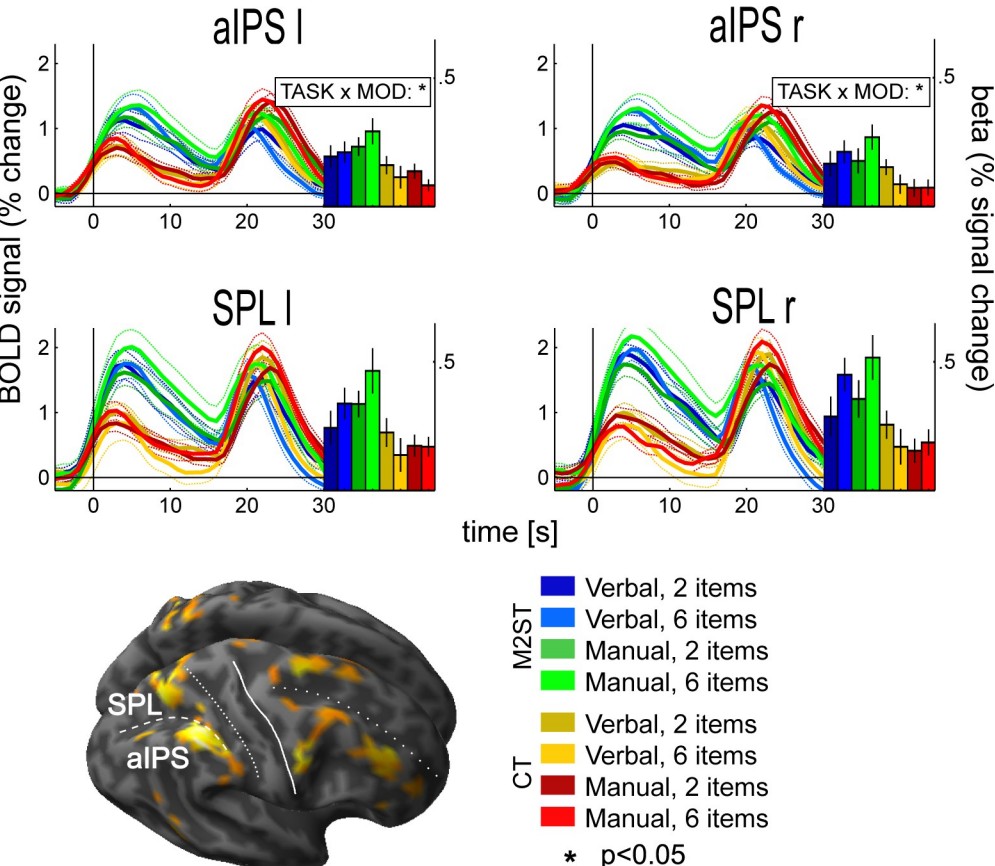

**Fig 5. WM activity as a function of response modality, load and task in parietal ROIs.** Individual bars reflect across-subject averages of delay-related beta estimates +/- SEM. Across-subject averages of fMRI-activity time-courses are shown in addition (+/- SEM [dashed lines]; vertical bars at 0s denote the onset of the delay phase of a trial). Blue and green colors denote verbal and manual response modalities of M2ST, respectively. Yellow and red colors denote verbal and manual responses in CT. Additional statistical information from a corresponding 2x2x2 repeated measures ANOVAs which was calculated across individual beta values considering the factors "Response Modality" (MOD: manual vs. verbal), "Load" (2 items vs. 6 items), and "Task" (match-to-sample task vs. control task) is provided (for details, please refer to RESULTS and Table 1). Significant signal differences between "verbal" and "manual" trials in M2ST as compared to CT (i.e. a significant interaction) were present in bilateral aIPS. This influence of response modality on maintenance-related brain activity was characterized by higher fMRI signal amplitudes in the M2ST whenever a hand response was prepared.

present in the control task, hence it is not merely attributable to "unspecific" effector preparation (or "motor readiness"). The major implication from these results is that the future motor context in which the memorized information is going to be used, has an influence on the neural processing of WM content. This principal finding offers a new perspective on the relationship between retrospective (sensory) and prospective (motor) processes that constitute working memory representations and suggests that WM memory representations do, whenever helpful, follow the sensory-to-motor transformation-chain for spatial content (Fig 1C).

It is important to emphasize, again, that our findings do not reflect a general, unspecific preparation of an effector or action [13, 25, 26], as this was controlled for by our control task. We also ensured through our task design that there is a clear separation between the memory maintenance and the preparation of any specific motor action throughout the delay period, a problem that has been extensively highlighted in other works [10, 11, 17]. This was because in our experiment an action could only be planned and executed after the delay when the test set

**Table 1. Results of ANOVA on fMRI activity in PPC.**

| ROI | Effect | df | F | p | p < .05 | Eta² |
|---|---|---|---|---|---|---|
| SPL left | load | 13 | 0.99 | 0.3383 | | 0.006 |
| | modality | 13 | 4.63 | 0.0509 | | 0.013 |
| | task | 13 | 15.16 | 0.0018 | * | 0.130 |
| | load:modality | 13 | 3.18 | 0.0977 | | 0.005 |
| | load:task | 13 | 6.38 | 0.0254 | * | 0.031 |
| | modality:task | 13 | 2.96 | 0.1092 | | 0.018 |
| | load:modality:task | 13 | 0.19 | 0.6688 | | 0.001 |
| SPL right | load | 13 | 3.7 | 0.07651 | | 0.019 |
| | modality | 13 | 0.25 | 0.6235 | | 0.001 |
| | task | 13 | 19.43 | 0.00071 | * | 0.160 |
| | load:modality | 13 | 2.62 | 0.12954 | | 0.004 |
| | load:task | 13 | 6.54 | 0.0239 | * | 0.037 |
| | modality:task | 13 | 2.39 | 0.14633 | | 0.013 |
| | load:modality:task | 13 | 0.7 | 0.41782 | | 0.004 |
| IPS left | load | 13 | 0.12 | 0.73011 | | 0.001 |
| | modality | 13 | 0.79 | 0.38936 | | 0.003 |
| | task | 13 | 23.25 | 0.00033 | * | 0.132 |
| | load:modality | 13 | 0.8 | 0.38836 | | 0.001 |
| | load:task | 13 | 8.58 | 0.01172 | * | 0.025 |
| | modality:task | 13 | 8.06 | 0.01395 | * | 0.024 |
| | load:modality:task | 13 | 0.45 | 0.51626 | | 0.002 |
| IPS left | load | 13 | 1.014 | 0.3324 | | 0.004 |
| | modality | 13 | 0.128 | 0.7262 | | 0.000 |
| | task | 13 | 11.577 | 0.0047 | * | 0.120 |
| | load:modality | 13 | 3.815 | 0.0726 | | 0.009 |
| | load:task | 13 | 12.599 | 0.0036 | * | 0.028 |
| | modality:task | 13 | 10.135 | 0.0072 | * | 0.018 |
| | load:modality:task | 13 | 0.098 | 0.7596 | | 0.000 |

was shown. This means that the observed activity differences did not simply reflect effector-specific action planning during the delay phase.

Given that this activity was neither related to the planning of a specific action nor to unspecific action preparation, we postulate that retrospective maintenance of working memory material is indeed modulated by the future motor context in which the material will be used. In Fig 6 we suggest a hypothetical organization of such a feedback-based modulation of WM through the motor context as is afforded by the stimulus material or, as in our experiment, a task rule. These "affordances" could allow the WM system to infer the "motor context" in which sensory information is likely to be used and to guarantee optimal storage in "working memory" for guiding future action. It is important to stress that our framework is distinct from the notion that object affordances could provide additional semantic clues to support the maintenance of objects in working memory [27, 28] as, in related studies, object affordances were usually independent of any upcoming behavioral response. Moreover, we'd like to highlight that our framework clearly extends the multicomponent model of working memory, as was laid out in the introduction, by providing hints that modality-specific storage is sensorimotor in nature. In addition, it seems to be rather in agreement with more recent "state-based models of working memory", which–according to D'Esposito and Postle [1]–"assume that the allocation of attention to internal representations—whether semantic LTM (e.g., letters, digits,

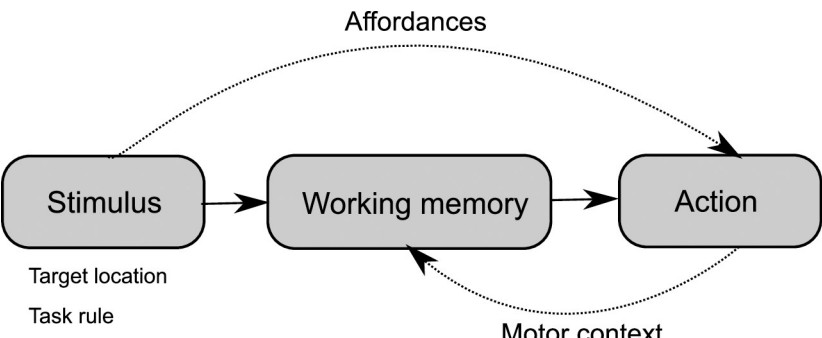

**Fig 6. Hypothetical (model) framework illustrating how the maintenance of mnemonic material could be modulated by future motor context.** Mnemonic stimulus material (e.g. visual target location, as in our experiment) and task rules afford the motor context in which this material is likely to be used. Predicted motor context information is fed back to WM processes to enable optimal storage for guiding future action.

words), sensory, or motoric—underlies the short-term retention of information in working memory" (page 3).

In the following part we will discuss our results in the context of previous research on the contributions of posterior parietal cortex to the maintenance of retrospective vs. prospective information. We will also derive practical consequences from our work, acknowledge potential limitations of our approach, and outline questions for future research.

Our findings extend earlier research investigating the roles of posterior parietal and prefrontal cortex in prospective and retrospective working memory. For example, Curtis et al. disentangled the retro- and prospective processes in a visuospatial WM task engaging saccadic eye movements. They showed that the frontal oculomotor areas maintain prospective WM contents when subsequent eye movements should be directed at the remembered spatial location [11]. In turn, posterior parietal areas rather processed retrospective information when the remembered location was not a target for a subsequent saccade. However, these authors could not exclude that their retrospective memory task–a non-match to sample task—required subjects to prospectively inhibit saccades to all remembered 'non-target' locations (which would be a type of prospective planning by itself). Following this argument, Lindner and colleagues [14], scrutinized the prospective (preparatory) role of PPC in planning: in a task that required both prospective and retrospective processing, PPC activity appeared to be driven mainly by prospective processes for both planning and inhibiting finger movements to visuospatial targets but, to a lesser extent, by retrospective memory, too. Our experiment surpasses these earlier findings by demonstrating a stronger involvement of posterior parietal cortex in visuospatial working memory maintenance whenever it involves the generation of a manual response (rather than a verbal one). As our task did not allow to plan (or inhibit) any specific motor response during the memory period, this effect does not relate to the planning of a specific action *per se*. Interestingly, the effect was the strongest (and only significant) in bilateral anterior IPS. The reason for this could be that our task required finger control (button presses) and, as opposed to more medial parietal areas, IPS is more specifically engaged in this particular type of action [18].This suggest a tight link between the sensory and the motor nature of WM.

Our results also have important practical implications: task-related brain activity in various working memory studies, which putatively tested maintenance of retrospective sensory information, might–at least in PPC–have been critically affected by the effector used for responding. This does not only call for a critical reconsideration of this earlier work. It also suggests careful task design in future studies–one that takes into account the various potential influences of behavioral (motor) components on working memory.

A limitation of our own study is that, so far, we have shown effector-dependent differences in WM maintenance only for spatial items and only in PPC, a region crucially involved in processing spatial information [11, 14, 21, 29]. On the basis of our current data we cannot determine whether a corresponding pattern of parietal activity, like the one reported here in parietal cortex for spatial content and manual responses, could be observed also for different memory modalities, e.g. the verbal content and verbal responses. It seems conceivable to think that patterns of activity would then generally shift towards areas representing the response effector that is congruent to the memorized material. For example, left lateral frontal cortex is specifically engaged both in the processing of verbal memory material [6] as well in the planning of verbal responses [22]. It is therefore likely that in this part of the brain one might likewise reveal an activity increase during the maintenance of verbal memory material. To scrutinize the detailed links between working memory and response preparation in other content and effector modalities than the ones presented here, more research is necessary.

In addition it would be interesting to see whether there is a true behavioral benefit from maintaining sensory information in a way that considers the relevant effector. The finding that eye- and hand- movements more affect the WM maintenance of spatial rather than verbal material may at least suggest that (spatial) WM maintenance depends on congruent sensorimotor associations which is consistent with our rationale [30]. Unfortunately, our paradigm used here was not designed to probe for detailed behavioral effects. To this end we would have to use the verbal and manual response modality not only in combination with visuospatial memory material, but also in combination with verbal material. Only then we would be able to control for unspecific performance differences (e.g. reaction times, capacity estimates, or hit rates) between both the two effectors and the two memory modalities.

Lastly, although we expected a higher activity for manual responses in parietal regions, visual inspection of our data suggests that the presence of such expected pattern could be more widespread, and span also across frontal motor ROIs (see S1 Fig). This would not necessarily be surprising and would still support our notion of a sensorimotor nature of WM representations, as these ROIs have been previously attributed to hand action planning and to the maintenance of visuospatial memory too [6, 14].

## Materials and methods

### Subjects

Fourteen subjects (two males; twelve females; mean age: 25) participated in this study. All of the participants had normal or corrected-to-normal visual acuity and were right-handed according to the Edinburgh Handedness Inventory [31]. None of them suffered from chronic, neurological or psychiatric diseases or took any medication. All subjects gave written informed consent prior to participation. Our experimental procedures were approved by the ethics committee at the University Clinic and the Medical Faculty of the University of Tübingen. Subjects received 10 Euro per hour for their participation. The group size was guided by power analyses performed on a similar, previously published fMRI dataset, comparing working memory and motor planning activity in posterior parietal cortex [14]. For a power of 0.80 and an alpha-level of 0.05 this analysis suggested a sample size of 11 subjects (two-tailed tests). To increase sensitivity, we initially decided to scan three further subjects. Please note that this sample size is larger or equal to studies performed in similar design [11, 14, 22, 32, 33, 34].

### Task

We applied functional magnetic resonance imaging (fMRI) while our participants worked on either a delayed match-to-sample task in which subjects had to memorize dot patterns or a

control task in which they had to judge if a given dot pattern is axially symmetrical (Fig 2). Trials belonging to both tasks were randomly interleaved within one functional run of the experiment. Each trial started with a baseline period (14,000 or 15,000ms), in which subjects were asked to keep central fixation on a fixation cross. Then, a cue (1000ms) indicated a) if the current trial was a memory trial (yellow square) or a control trial (blue square) and b) if subjects would have to respond verbally (picture of a mouth) or manually (picture of a hand) in the end of the trial. In the memory trials, this cue period was followed by an encoding period (3000ms), in which subjects saw 18 small circles that were arranged in a circle around the central cue. Either two or six of these circles were filled, and their relative positions within the big circular arrangement served as the spatial memory items that subjects had to remember. Our spatial memory task design deliberately refrained from a sequential presentation of these visuospatial items. This was to avoid item ordering, which could affect WM processes of encoding and recall and, importantly, respective activity modulations in parietal regions that we could not easily control for [35]. The encoding screen was subsequently masked by a scrambled dot-pattern for 500ms to prevent afterimages of the cue and encoding screen. Then the delay epoch began (14,000 or 15,000ms). Afterwards, we either presented the very same dot pattern of the encoding period or a different one (3000ms). Subjects had to wait until they saw a green go-cue and had to indicate within 3000ms if this dot pattern did or did not match the one of the encoding period by either saying 'same' or 'different', respectively in the verbal conditions, or by pressing the right or left button of a button box, respectively, in the manual conditions. The control condition differed from the memory condition after the cue period insofar, as a circular arrangement consisting of 18 merely unfilled (but no filled) circles was presented, so that subjects would not have to maintain any memory content during the subsequent delay period (14,000ms or 15,000ms). Then, we presented a circle composed of 18 small circles after the delay period. Again, either two or six of these circles were filled. Subjects waited until the green "go" cue and indicated whether or not this dot pattern was axially symmetrical to the vertical midline of the big circle by either saying "same" or "different", respectively, in the verbal conditions or by pressing the right or left button of a button box, respectively, in the manual conditions. Lastly, a blank screen was displayed for the remaining 500ms of the trial. Our subjects worked on five consecutive blocks in which each of the 4 main conditions, i.e. 2 modalities (verbal vs. manual) x 2 tasks (memory vs. control), was presented twice and in a pseudo-randomized order, resulting in a total of 20 trials per condition for our main comparison M2ST vs. CT. Half of these trials (N = 10) comprised of 2 items, the other half (N = 10) comprised of 6 items.

## Stimulus presentation

We created the visual stimuli on a Windows™ based PC using MATLAB R2007b (The Math-Works, Inc.) and Cogent Graphics developed by John Romaya at the LON at the Wellcome Department of Imaging Neuroscience. They were projected onto a translucent screen (size of the projected image: 28 deg x 37 deg visual angle; viewing distance: 92 cm) by means of a video projector (frame rate: 60 Hz; resolution: 1024 x 768 pixels). Our participants watched the projected stimuli on the translucent screen being placed behind them with the aid of a mirror that was mounted on the head coil.

We displayed the fixation cross of the baseline and delay phases in Arial font and a 2.44 degrees visual angle font size. The squared color cue was 2.44 deg x 2.44 deg. The modality cue images were 1.95deg x 1.76deg. The spatial cues (dots) were placed at 4.9 deg radius from the central fixation point and were approximately 1 degree in diameter, each.

## Data acquisition

**Eye tracking.** Our subjects were supposed to maintain central fixation during the whole trial (besides the response phases) to ensure that fMRI activity would not be influenced by eye movements. We recorded eye-movements with a MRI-compatible infrared eye-camera (SMI SensoMotoric Instruments) and the ViewPoint Eye Tracker system (Arrington Research Inc.; sampling rate: 50 Hz) and performed on-line visual inspection of the eye image to ensure that subjects adhered to the fixation instruction. The proper detection of fixational saccades during scanning was not possible due to camera noise.

**WM performance.** In the verbal conditions, we recorded our subjects' answers with a MRI-compatible microphone (Optoacoustics Dual-Channel Microphone, Optoacoustics Ltd., Israel; sampling rate: 8 kHz). The manual responses were recorded with a button box with two buttons indicating either "same" (left button) or "different" (right button). All recordings were analyzed off-line using self-written scripts in MATLAB R2007b (The MathWorks, Inc.).

**fMRI data acquisition.** We collected the MR images with a 3-Tesla MR-scanner and a twelve channel head coil (Siemens TRIO, Erlangen, Germany). A T1-weighted magnetization-prepared rapid-acquisition gradient echo (MP-RAGE) structural scan of the whole brain was assessed from each subject (number of slices: 176, slice thickness: 1mm, gap size: 0 mm, in-plane voxel size: 1 x 1 mm, TR: 2300 ms, TE: 2.92 ms, FOV: 256 x 256 mm, resolution: 256 x 256 voxels). Moreover, we acquired T2*-weighted gradient-echo planar imaging (EPI) scans (slice thickness: 3.2 mm, gap size: 0.8 mm, in-plane voxel size: 3 x 3 mm, TR: 2000 ms, TE: 30 ms, flip angle: 90˚, FOV: 192 x 192 mm, resolution: 64 x 64 voxels, 32 axial slices). 330 EPIs were collected from each participant during five consecutive runs of 11 minutes each. Cerebral cortex and most sub-cortical structures were completely covered by the EPI-volume but we did not record from the most posterior parts of the cerebellum in several of our subjects due to brain size.

## Behavioral performance analysis

We statistically analyzed our behavioral data using SPSS (IBM SPSS Statistics, version 22) and R (R Foundation for Statistical Computing). Furthermore, functional MRI data were analyzed using SPM8 (Wellcome Department of Cognitive Neurology, London, UK) and R (R Foundation for Statistical Computing).

To investigate if the performance level was influenced by the modality, the task and/ or the load level, we analyzed share of correct answers by means of a three-way repeated measures ANOVA with the factors 'response modality' (2 levels: verbal vs. manual), 'task' (2 levels: memory vs. control), and 'load' (2 levels: 2 vs. 6).

## fMRI data analysis

Pre-processing. The pre-processing of our functional images was done in SPM8 (Wellcome Department of Cognitive Neurology, London, UK). Separately for each subject, we realigned all functional images by using the first scan of the first session as a reference. Then, we spatially coregistered the T1 anatomical image to the mean image of the functional scans and normalized our subjects' mean anatomical images to the SPM T1 template in MNI space (Montreal Neurological Institute). The resulting normalization parameters were also applied to all functional images for spatial normalization. Finally, all functional images were smoothed by using a Gaussian filter (6 x 6 x 8 mm$^3$ full-width at half-maximum) and high-pass filtered (cutoff period: 100 ms).

**First-level analysis.** In the subject-level fMRI analysis we specified a GLM in which each of our eight conditions (2 tasks [M2ST & CT] x 2 response modalities [verbal & manual] x 2

'loads' [2 and 6 circles]) was modeled as a separate boxcar regressor of respective duration for both the delay period as well as for the response phase. We modelled the cue-period (+mask) accordingly but only considered two regressors representing our two principle tasks (M2ST & CT). All aforementioned regressors were convolved with the canonical haemodynamic response function in SPM8. Fixation epochs weren't modeled explicitly and served in the model as an implicit baseline. Head motion estimates that were assessed during realignment were included in the GLM as six independent regressors (three regressors for head rotation and translation, each).

**Group-level analysis.** Group-level activity maps were plotted on the basis of a $2^{nd}$ level random-effects analysis from the first level statistical parametric maps of activity related to the delay regressors in the CT and the M2ST (M2ST>CT). We thresholded statistical parametric maps at p<0.001, uncorrected for multiple comparisons, to ensure that our exploratory analysis of the whole brain does not omit any substantial activity during the delay phase. The resulting parametric maps were then overlayed on the standard MNI T1 template image as provided by SPM8 in order to anatomically define the regions of activity. Note that we used these maps only for visualization and aiding in selecting group-based coordinates for subsequent ROI selection in some individual subjects.

**ROI analysis.** We focused the analyses on comparing activity in cortical areas involved in our working memory task and, in particular, areas in PPC (compare introduction). For the analysis of signal amplitudes, we selected areas in each individual subject, namely those that showed a statistically significant increase in BOLD intensity during the delay phase of working memory trials (as compared to control trials). We applied a statistical significance threshold of p<0.05, corrected for family-wise error and only report clusters of 10 or more adjacent voxels. The maps were then overlaid on each subject's T1 images, thus allowing precise assessment of the anatomical location of WM-related activation. Whenever this threshold did not yield any clusters of activation in a given subject in this full-brain analysis, we instead applied a small-volume correction centering on the coordinates of planning and spatial memory regions as were described by our group analyses (see S1 Table). Small-volume correction was performed for a radius of 20mm around the respective ROI coordinates (p<0.05, FWE-corrected within this volume sphere). In one of the subjects, where the small-volume correction did not yield any significant clusters of activity, we used a t contrast comparing delay-related M2ST-activity against the implicit baseline (p<0.05, FWE-corrected). As we did not differentiate between the "response modality" at this stage of functional ROI definition (i.e. the activity maps were calculated for all conditions, pooled across both response modalities), our approach ensured that our ROI selection was not biased in favor of our hypothesis [36].

The ROIs were defined by the location of their corresponding local maxima of t values within major spatial clusters of working memory related BOLD activity. In particular, these were: left and right superior parietal lobule (SPL) and left and right anterior intraparietal sulcus (aIPS). In addition we considered frontal ROIs: left and right dorsal premotor cortex (PMd), left and right ventral premotor cortex (PMv), left and right dorsolateral prefrontal cortex (DLPFC), and supplementary motor area (SMA) due to their engagement in WM.

## Conclusions

In summary, our results suggest that the BOLD activity underlying working memory tasks is affected by the future motor context, namely the instructed effector. This effect was prominent in posterior parietal cortex, repeatedly demonstrated to be involved in both the maintenance of visuospatial information and the prospective planning of actions. Whether the posterior parietal cortex is indeed the key area bridging between the retrospectively processed

mnemonic content and its future use (whether in laboratory or natural world) remains to be answered. In any case, our findings shed a new light on the actual structure of the working memory network and its intimate link to brain systems engaged in sensorimotor transformations for behavior.

## Supporting information

**S1 Data.**
(CSV)

**S2 Data.**
(CSV)

**S1 Table. Average MNI coordinates of ROIs, +/- standard deviation.**
(DOC)

**S2 Table. The table shows the results of our 2x2 repeated measures ANOVAs with the factors "task" (M2ST cs CT), "response modality" (manual vs. verbal) and "load" (2 vs. 6 items), calculated across delay-related betas extracted from non-parietal ROIs.**
(DOC)

**S1 Fig. WM-related brain activity in non-parietal ROIs.** Individual bars reflect across-subject averages of delay-related beta estimates +/- SEM in M2ST. Across-subject averages of fMRI-activity time-courses are shown in addition (+/- SEM [dashed lines]; vertical bars at 0s denote the onset of the delay phase of a trial). Blue and green colors denote verbal and manual response modalities of M2ST, respectively. Yellow and red colors denote verbal and manual responses in CT. Additional statistical information from a corresponding 2x2x2 repeated measures ANOVAs which was calculated across individual beta values considering the factors "Response Modality" (MOD: manual vs. verbal), "Load" (2 items vs. 6 items), and "Task" (match-to-sample task vs. control task) is provided in S2 Table.
(DOC)

## Author Contributions

**Conceptualization:** Artur Pilacinski, Melanie S. Höller-Wallscheid, Axel Lindner.

**Data curation:** Artur Pilacinski, Melanie S. Höller-Wallscheid, Axel Lindner.

**Formal analysis:** Artur Pilacinski, Axel Lindner.

**Funding acquisition:** Axel Lindner.

**Investigation:** Artur Pilacinski, Melanie S. Höller-Wallscheid, Axel Lindner.

**Methodology:** Artur Pilacinski, Melanie S. Höller-Wallscheid, Axel Lindner.

**Project administration:** Artur Pilacinski, Axel Lindner.

**Resources:** Artur Pilacinski, Axel Lindner.

**Software:** Artur Pilacinski, Axel Lindner.

**Supervision:** Artur Pilacinski, Axel Lindner.

**Validation:** Artur Pilacinski, Axel Lindner.

**Visualization:** Artur Pilacinski, Axel Lindner.

**Writing – original draft:** Artur Pilacinski, Axel Lindner.

**Writing – review & editing:** Artur Pilacinski, Axel Lindner.

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
