## [Decision Letter · Decision Letter 0]

15 Jun 2020

PONE-D-20-06230

Effector-dependent working memory maintenance

in posterior parietal cortex

PLOS ONE

Dear Dr. Pilacinski,

Thank you for submitting your manuscript to PLOS ONE. After careful consideration, we feel that it has merit but does not fully meet PLOS ONE’s publication criteria as it currently stands. Therefore, we invite you to submit a revised version of the manuscript that addresses the points raised during the review process.

We look forward to receiving your revised manuscript.

Kind regards,

Kiyoshi Nakahara, PhD

Academic Editor

PLOS ONE

Journal Requirements:

Reviewers' comments:

Reviewer's Responses to Questions

**Comments to the Author**

1. Is the manuscript technically sound, and do the data support the conclusions?

Reviewer #1: Partly

Reviewer #2: Yes

2. Has the statistical analysis been performed appropriately and rigorously? 

Reviewer #1: No

Reviewer #2: No

3. Have the authors made all data underlying the findings in their manuscript fully available?

Reviewer #1: No

Reviewer #2: Yes

4. Is the manuscript presented in an intelligible fashion and written in standard English?

Reviewer #1: Yes

Reviewer #2: Yes

5. Review Comments to the Author

Reviewer #1: This manuscript reports on an fMRI study of visual working memory. The main aim of the study appears to have been to elucidate the nature of “Effector-dependent working memory” neural mechanisms. What exactly this means is not clear, nor is it clear how the experimental design and reported findings address this aim. Furthermore, in my view, possible experimental confounders limit the impact of the results.

Major concerns:

I have to admit that I have a hard time grasping the story of the manuscript. A large number of previous articles have been cited in the introduction, while it lacked a sufficient theoretical framework to reach their hypothesis. I would suggest that the authors would try to frame their findings from a theoretical perspective. Furthermore, in my view, it may not be a good choice to add a figure in the introduction section for a research article.

The methods and results sections are very unstructured and confusing. The major issue is that the simple size is small. While the authors have argued that they choose the sample size based on a previous fMRI study, and even say that their sample size is larger than several fMRI studies, more than 20 participants is the norm now. Further, the voxel-wise threshold of .01 is considered liberal these days (see Woo et al., 2014, NeuroImage 91:412-419). Additionally, there appears to be no correction method applied? eg, cluster correction, etc. At least the same participants' information has been repeated in the results section (pp.6 - 7), also a part of fMRI data analyzing also repeated.

The discussion is unclear, superficial and difficult to follow. There are also some points that are incorrect or incomplete and often deriving.

In sum, the research topic of the paper is somewhat interesting, but the realization, experimental design and analysis raise many open questions (I have only mentioned a few above) that severely limit the interpretability and generalization of the results. Therefore, I cannot support the publication of this paper.

Reviewer #2: The authors describe the involvement of Posterior Parietal Cortex (PPC) during maintenance of a visuospatial information where a response has to be done particularly with a manual (hand) selection. The study combines behavioral experiment and functional neuroimaging involving healthy subjects. In the experiment, subjects were presented with a set of memory items in the form of filled/empty circles and asked to indicate whether the test set presented after matched the memory set. At the start of each trial, subjects were also given a cue whether the trial response had to be done verbally or manually. The requirement to sample-and-match was not present in a control condition where subjects had to simply indicate whether the test stimulus was symmetrical. This work is an extension of earlier work dealing with prospective and retrospective WM, and the involvement of PPC in it (e.g. Linder et al, 2010).

While the study design is simple and findings are interesting, there are a few points that need to be addressed:

1. In Task section (line 357) under Materials and Methods, the statement is unclear whether each subject had both tasks that were interleaved at random, or subjects were randomly assigned to one of the groups (control vs experimental group). This should be clarified in the main text. If the latter is used, then a 3-way repeated measures ANOVA is not correct, and a mixed-design analysis may be used.

2. If a repeated measures ANOVA was used in the statistical analyses, was the test of sphericity/Greenhouse-Geisser correction performed here?

3. In the manual task, subjects responded manually by pressing the left button for ‘same’, and the right button for ‘different’ (as stated in line 384-387). Was the same instruction given for the control task (left button for ‘same/ symmetrical’, and right for ‘different/ asymmetrical’)?

4. The main finding of the study is on the maintenance-related brain activities that are dependent on the type of end-effector. Does the use of different end-effector affect the working memory performance itself? If not, perhaps the word “influence” in the statement “the future motor context ..., has an influence on memory maintenance” (Line 225) needs clarification. It may also be emphasized that Figure 6 refers to the hypothetical WM model deduced from the neural activities, not from behavioral outcomes.

Additional minor typos/errors:

Line 65, pg3

maintenance of retrospective vs. prospective information...

Line 75, pg4

targeted for an upcoming...

Line 381, pg16

presented , subjects would not... (add a comma instead)

6. PLOS authors have the option to publish the peer review history of their article (what does this mean?). If published, this will include your full peer review and any attached files.

Reviewer #1: No

Reviewer #2: No

---

## [Author Response · Author response to Decision Letter 0]

6 Jul 2020

Please find enclosed our revised manuscript titled “Remember how to use it: effector-dependent modulation of spatial working memory activity in posterior parietal cortex” (changed from “Effector-dependent working memory maintenance in posterior parietal cortex). Note we changed the title of the manuscript to better reflect the topic and content of our manuscript. We addressed all the editorial comments and as per the editorial request and we now attach the relevant data as Supplementary Material to fulfill the journal’s policy. Moreover, we updated our funding acknowledgments listing funding sources and types.

We want to thank both reviewers for their constructive comments. The attached manuscript contains now all suggested changes and we also tried hard to advance the text flow to improve readability. All changes to the manuscript have been highlighted in yellow. Below we attach our reply to reviewers’ specific comments. We believe we managed to address all issues brought up during the review process through our changes to the manuscript and/or our clarifications given below.

Reviewer #1: This manuscript reports on an fMRI study of visual working memory. The main aim of the study appears to have been to elucidate the nature of “Effector-dependent working memory” neural mechanisms. What exactly this means is not clear, nor is it clear how the experimental design and reported findings address this aim. Furthermore, in my view, possible experimental confounders limit the impact of the results.

Major concerns:

I have to admit that I have a hard time grasping the story of the manuscript. A large number of previous articles have been cited in the introduction, while it lacked a sufficient theoretical framework to reach their hypothesis. I would suggest that the authors would try to frame their findings from a theoretical perspective. Furthermore, in my view, it may not be a good choice to add a figure in the introduction section for a research article.

The notion of effector-dependent working memory maintenance (not “Effector-dependent working memory”) stems from our findings, bridging between the established fields of motor planning (see e.g. Curtis et al., 2004; Lindner et al., 2010) and (spatial) working memory (see e.g. Curtis & D’Esposito, 2006; Buchsbaum & D’Esposito, 2019; Myerson et al., 1999), as we show the direct interplay between these two processes and their shared neural components. We laid this out in more detail in the introduction and discussion and their corresponding model figures. We have now improved the theoretical parts of the manuscript to make the reader better understand the concepts we used.

The methods and results sections are very unstructured and confusing. The major issue is that the simple size is small. While the authors have argued that they choose the sample size based on a previous fMRI study, and even say that their sample size is larger than several fMRI studies, more than 20 participants is the norm now. 

We generally share the concern about reproducibility of results in neuroscience and other disciplines. For precisely this reason, when designing our study, we grounded our selection of sample size on previous research and, more importantly, on a power analysis (compare page 15 of the manuscript). We believe this is in accordance with good scientific practice (see e.g. Poldrack etal., 2017 for a methodological review). We used this power analysis to guide our sample size estimate, yet still increased number of trials and sample size compared to the study we based upon (Lindner et. al., 2010) in order to improve power of our design, within feasible boundaries. 

However, we must emphasize that most published studies from leading researchers in the field, utilizing similar ROI analyses in event-related fMRI-designs with comparably long delay- (and inter-trial-) periods and a similar number of experimental conditions, have recruited equal or even lower numbers of subjects (see e.g.: Gallivan et al., 2011, J. Neurosci., N=8; Gallivan et al., 2014, Curr. Biol., N=13; Gallivan et al., 2013, eLife, N=13; Gallivan et al., 2019, Cereb. Cortex, N=14; Curtis & Connoly, 2007, J. Neurosci., N=13; Tartk & Curtis, 2009, Nat. Neurosci., N=13; Medendorp et al., 2006, J. Neurophys., N=7; Kadmon Harpaz et al., 2014, Neuron, N=11; Pitzalis et al., 2016, Neuroimage, N=12; Beurze et al., 2010, J. Neurophys., N=14). These sample sizes seemingly yield consistent and stable results within the ROI approach we also use.

Further, the voxel-wise threshold of .01 is considered liberal these days (see Woo et al., 2014, NeuroImage 91:412-419). Additionally, there appears to be no correction method applied? eg, cluster correction, etc. At least the same participants' information has been repeated in the results section (pp.6 - 7), also a part of fMRI data analyzing also repeated.

First off, please note that we used threshold of p<0.001 (not p<0.01) only to obtain a group map of well-described functional regions in order to provide a general overview of the set of areas engaged in our WM task and not for any statistical inferences testing our hypothesis. We specifically write that “we used these maps only for visualization and aiding in selecting group-based coordinates for subsequent ROI selection in some individual subjects” (lines 543-545).

For ROI selection in individual subjects “we applied a statistical significance threshold of p<0.05, corrected for family-wise error” (lines 551-552).

The discussion is unclear, superficial and difficult to follow. There are also some points that are incorrect or incomplete and often deriving.

As this point of the review is formulated in somewhat general terms we could see no way of satisfying the reviewer’s need for clarity other than going through the manuscript again in order to correct any points we found potentially unclear or confusing for the reader. For this we maintained the manuscript layout and text and have polished the text flow where we found appropriate (all changes highlighted in yellow). We believe that the changes we implemented did improve the clarity of the text.

Reviewer #2: The authors describe the involvement of Posterior Parietal Cortex (PPC) during maintenance of a visuospatial information where a response has to be done particularly with a manual (hand) selection. The study combines behavioral experiment and functional neuroimaging involving healthy subjects. In the experiment, subjects were presented with a set of memory items in the form of filled/empty circles and asked to indicate whether the test set presented after matched the memory set. At the start of each trial, subjects were also given a cue whether the trial response had to be done verbally or manually. The requirement to sample-and-match was not present in a control condition where subjects had to simply indicate whether the test stimulus was symmetrical. This work is an extension of earlier work dealing with prospective and retrospective WM, and the involvement of PPC in it (e.g. Linder et al, 2010).

We first want to thank again the Reviewer #2 for the thorough and helpful review and especially for pointing out issues that helped improving our manuscript. Below we list the Reviewer’s comments with our replies.

While the study design is simple and findings are interesting, there are a few points that need to be addressed:

1. In Task section (line 357) under Materials and Methods, the statement is unclear whether each subject had both tasks that were interleaved at random, or subjects were randomly assigned to one of the groups (control vs experimental group). This should be clarified in the main text. If the latter is used, then a 3-way repeated measures ANOVA is not correct, and a mixed-design analysis may be used.

We used repeated measures ANOVA and trials belonging to both experimental condition were randomly interleaved within an experimental run. We now include this information at line 433 of the manuscript for clarity. Thank you for noticing this. 

2. If a repeated measures ANOVA was used in the statistical analyses, was the test of sphericity/Greenhouse-Geisser correction performed here?

As we used the 2x2x2 design in our analyses, there was no need for including sphericity correction as in that case (only two levels per factor) the sphericity assumption will always be met.

3. In the manual task, subjects responded manually by pressing the left button for ‘same’, and the right button for ‘different’ (as stated in line 384-387). Was the same instruction given for the control task (left button for ‘same/ symmetrical’, and right for ‘different/ asymmetrical’)?

Yes, we had the same responses in both tasks. Thank you for pointing this out, we now include this information in lines 138-142 of the manuscript.

4. The main finding of the study is on the maintenance-related brain activities that are dependent on the type of end-effector. Does the use of different end-effector affect the working memory performance itself? If not, perhaps the word “influence” in the statement “the future motor context ..., has an influence on memory maintenance” (Line 225) needs clarification. It may also be emphasized that Figure 6 refers to the hypothetical WM model deduced from the neural activities, not from behavioral outcomes.

That’s right. We now modified this and the following passage to better reflect our point that the predicted context influences neural processing of sensorimotor information maintained in working memory (lines 284-289).

Additional minor typos/errors:

Line 65, pg3

maintenance of retrospective vs. prospective information...

Line 75, pg4

targeted for an upcoming...

Line 381, pg16

presented , subjects would not... (add a comma instead)

Thank you for pointing these errors out. We corrected them accordingly.

---

## [Decision Letter · Decision Letter 1]

10 Aug 2020

Remember how to use it: effector-dependent modulation of spatial working memory activity in posterior parietal cortex

PONE-D-20-06230R1

Dear Dr. Pilacinski,

We’re pleased to inform you that your manuscript has been judged scientifically suitable for publication and will be formally accepted for publication once it meets all outstanding technical requirements.

Kind regards,

Kiyoshi Nakahara, PhD

Academic Editor

PLOS ONE

Additional Editor Comments (optional):

Reviewers' comments:

Reviewer's Responses to Questions

**Comments to the Author**

1. If the authors have adequately addressed your comments raised in a previous round of review and you feel that this manuscript is now acceptable for publication, you may indicate that here to bypass the “Comments to the Author” section, enter your conflict of interest statement in the “Confidential to Editor” section, and submit your "Accept" recommendation.

Reviewer #1: (No Response)

Reviewer #2: All comments have been addressed

2. Is the manuscript technically sound, and do the data support the conclusions?

Reviewer #1: (No Response)

Reviewer #2: Yes

3. Has the statistical analysis been performed appropriately and rigorously? 

Reviewer #1: (No Response)

Reviewer #2: Yes

4. Have the authors made all data underlying the findings in their manuscript fully available?

Reviewer #1: (No Response)

Reviewer #2: Yes

5. Is the manuscript presented in an intelligible fashion and written in standard English?

Reviewer #1: (No Response)

Reviewer #2: Yes

6. Review Comments to the Author

Reviewer #1: (No Response)

Reviewer #2: (No Response)

7. PLOS authors have the option to publish the peer review history of their article (what does this mean?). If published, this will include your full peer review and any attached files.

Reviewer #1: No

Reviewer #2: No